# A Review of the Green Synthesis of ZnO Nanoparticles Utilising Southern African Indigenous Medicinal Plants

**DOI:** 10.3390/nano12193456

**Published:** 2022-10-03

**Authors:** Dorcas Mutukwa, Raymond Taziwa, Lindiwe Eudora Khotseng

**Affiliations:** 1Department of Chemistry, University of the Western Cape, Robert Sobukwe Rd., Private Bag X17, Bellville 7535, South Africa; 2Department of Applied Science, Faculty of Science Engineering and Technology, Walter Sisulu University, Old King William Town Road, Potsdam Site, East London 5200, South Africa

**Keywords:** green synthesis, zinc oxide nanoparticles, plant-mediated synthesis, medicinal plants nanoparticles

## Abstract

Metal oxide nanoparticles (NPs), such as zinc oxide (ZnO), have been researched extensively for applications in biotechnology, photovoltaics, photocatalysis, sensors, cosmetics, and pharmaceuticals due to their unique properties at the nanoscale. ZnO NPs have been fabricated using conventional physical and chemical processes, but these techniques are limited due to the use of hazardous chemicals that are bad for the environment and high energy consumption. Plant-mediated synthesis of ZnO NPs has piqued the interest of researchers owing to secondary metabolites found in plants that can reduce Zn precursors and stabilise ZnO NPs. Thus, plant-mediated synthesis of ZnO NPs has become one of the alternative green synthesis routes for the fabrication of ZnO NPs. This is attributable to its environmental friendliness, simplicity, and the potential for industrial-scale expansion. Southern Africa is home to a large and diverse indigenous medicinal plant population. However, the use of these indigenous medicinal plants for the preparation of ZnO NPs is understudied. This review looks at the indigenous medicinal plants of southern Africa that have been used to synthesise ZnO NPs for a variety of applications. In conclusion, there is a need for more exploration of southern African indigenous plants for green synthesis of ZnO NPs.

## 1. Introduction

Nanomaterials (NMs) have gained prominence in the 21st century owing to their ability and flexibility in addressing global research problems in fields of energy [1], water and water sanitation [2], medicine [3], agriculture [4], material science [5], and cosmetic industry [6]. In 1959, Richard Feynman’s phrase [7], “There’s plenty of Room at the Bottom” emerged as a versatile and flexible platform capable of delivering efficient, cost-effective, and environmentally friendly solutions to humanity’s sustainability challenges.

The term NMs refers to materials with a diameter ranging between 1 and 100 nm. NMs have attained prominence in technological advancement as a result of their unique flexible and adaptable optical, mechanical, electrical, biological catalytic, and magnetic characteristics that are superior to their bulk form [8,9]. The large surface area and size of NMs result in their unique properties, which are tuneable to suit their application. Furthermore, the distinct size, shape, and structure of NMs influence their reactivity, toughness, and other properties. These properties make them good candidates for a wide range of industrial and household uses, including environmental, imaging, medical, energy, catalysis, cosmetics, and medical applications [10].

Metal oxide nanoparticles (NPs) such as copper oxide (CuO), tin oxide (SnO_2_), zinc oxide (ZnO), and titanium dioxide (TiO_2_) are among the most studied NMs due to their numerous diverse properties and applications. Zinc oxide is one such metal oxide that has received extensive research due to its chemical stability, non-toxic nature, cost-effectiveness, biocompatibility, and biodegradable nature [11,12]. To date, a number of different chemical and physical synthesis methods, including laser ablation, vapour deposition, electrodeposition, sol-gel, spray pyrolysis, hydrothermal, and microwave-assisted methods, have been employed to prepare ZnO NPs. However, some of these synthesis methods are hampered by high costs and the usage of hazardous chemicals and solvents that are harmful to the environment, as well as by high energy consumption. [11,13]. As a result, new synthesis methods for the fabrication of ZnO NPs that are both safe and cost-effective are required. Thus, attention has been drawn toward green/biological synthesis as a substitute to conventional chemical and physical synthesis methods.

Plants, bacteria, yeast, and algae have successfully been used to synthesise ZnO NPs, with plants showing great potential. This is due to plant-mediated synthesis being easily scalable for industrial fabrication, simple, environmentally friendly, cost-effective, and time-effective. Plants are known to possess secondary metabolites known as phytochemicals, which include phenols, saponins, tannins, alkaloids, flavonoids, and carbohydrates that can be used as alternatives to toxic organic and inorganic chemicals, which are utilised as reducing or stabilising agents in conventional preparation of ZnO NPs [14,15].

*Cassia fistula* [16], *Euphorbia hirta* [17], *Rosa indica* [18], *Costus pictus* [19], and *Solanum nigrum* [20] are medicinal plants that have been successfully employed in the fabrication of ZnO NPs. South Africa, found in southern Africa, is not only home to a unique and diverse plant species, with some of these species having medicinal properties, but it is also mostly endemic [21]. However, there have been few reports on the use of traditional medicinal plants from southern Africa in the preparation of ZnO NPs. As a result, there is a need to investigate more medicinal plants for the preparation of ZnO NPs.

*Agathosma betulina* [22], *Plumbago auriculata* [23], *Monsomia burkeana* [24], *Lessertia montana* [25], *Lessertia frutescens* [26], *Tulbaghia violacea* [27], *Aspalathus linearis* [28], *Dovyalis caffra* [29,30], and *Athrixia phylicoides* DC [31] are some of the traditional medicinal plants that are endemic to South Africa and southern Africa that have been investigated for the fabrication of ZnO NPs and the as-synthesised ZnO NPs explored for various applications. This review will focus on some medicinal plants indigenous to South Africa and parts of southern Africa that have been explored for the preparation of ZnO NPs. It will focus on the medicinal plant overview, characterisation, and applications of the synthesised ZnO NPs. We believe that this review will draw more attention to the green synthesis of ZnO NPs using medicinal plants and lead to more exploration of the vast diversity of plant species in southern Africa.

## 2. Properties and Synthesis of ZnO NPs

### 2.1. Properties of ZnO NPs

The hexagonal wurtzite, cubic zinc blende, and rock salt shown in Figure 1 are the three known crystalline phases of ZnO. ZnO is commonly found in the wurtzite phase because it is stable under ambient conditions. Both the rock salt and the zinc blende are cubic and unstable under ambient conditions. Growing the zinc blende on cubic substrates makes it stable, whereas the rock salt phase can be obtained by converting ZnO under relatively high pressure [32]. The wurtzite phase belongs to the P6_3_mc space group and has a hexagonal geometry with two interconnecting sublattices of Zn^2+^ and O^2−^ such that each Zn ion is surrounded by four tetrahedral O^2−^ ions and vice-versa. This geometry lacks inversion symmetry along the hexagonal axis giving rise to a polarity symmetry and resulting in piezoelectric and spontaneous polarisation properties [33].

ZnO NP is a low-cost n-type bandgap semiconductor that is dielectric, transparent, and piezoelectric. It has a wide bandgap of 3.37 eV at room temperature (RT), large exciton binding energy of 60 meV and high thermal conductivity. This allows its application in optoelectronics, gas sensors, photocatalysis, aerospace, light-emitting diodes, and photovoltaics. ZnO NPs are biocompatible, anti-inflammatory, anticancer, and have wound healing properties, which make them suitable for biomedical applications [34,35].

### 2.2. Synthesis of ZnO NPs

An ideal synthesis method for the preparation of NPs needs to be low-cost, eco-friendly, able to produce high-quality NPs of desired morphology and size to suit applications of the NPs. Physical, chemical, and biological synthesis methods have been employed in the preparation of NPs, and these can be classified into top-down and bottom-up approaches, as illustrated in Figure 2. The top-down approach includes both physical and chemical synthesis methods, which generally involve using chemicals or force to break down bulk material into smaller particles. The top-down approach methods include ball milling, laser ablation, chemical etching, laser pyrolysis, mechanochemical, solid state, etc. [11,36].

The chemical synthesis methods that fall under the bottom-up approach include sol-gel, hydrothermal, microwave irradiation, spray pyrolysis, chemical precipitation, solvochemical, microemulsion, etc. This approach involves nucleation of ions, atoms or molecules in solution followed by aggregation of the nanoparticles. It is the approach that has been extensively reported for the preparation of NPs. Both physical and chemical conventional synthesis processes mentioned above have been reported for the preparation of ZnO NPs. However, some of these conventional synthesis routes are limited by high energy consumption and high pressure and require complex equipment, resulting in a high overall cost [37,38]. Moreover, these methods can also be limited by toxic chemicals used during synthesis, which can be harmful to the environment and the person handling the chemicals. These toxic chemicals are mostly used as reducing agents or capping agents and include sodium borohydride, hydrazine, ethylene glycol, dimethylformamide, cetyl trimethylammonium bromide (CTAB), pyridine, polyethylene glycol, etc. [39]. Therefore, safer and cost-effective synthesis methods are needed for the preparation of NPs. Thus, green synthesis using microorganisms or plant extracts can be a substitute to conventional chemical and physical synthesis methods.

### 2.3. Green Synthesis

Biological synthesis is a bottom-up approach that utilises plants, fungi, bacteria, algae, etc., for the preparation of NPs. This green synthesis approach eliminates the need for reducing and capping agents during the synthesis of NPs. This is due to the ability of microbes and plants to reduce metal salt precursors while also stabilising the synthesised NPs using naturally occurring substances such as enzymes and secondary metabolites. Thus, making biological synthesis a safer and low-cost alternative to conventional NPs synthesis methods.

Suba and associates [40] successfully synthesised ZnO NPs using bacteria isolated from cow milk, and the synthesised ZnO NPs exhibited good antibacterial and anticancer activity. The ZnO NPs were hexagonal in structure and spherical in shape and the average particle size was reported to be 32 nm. Additionally, Hefny et al. [41] successfully synthesised ZnO NPs using five fungal cultures isolated from five fungal species (*Aspergillus niger*, *Aspergillus tubulin*, *Aspergillus fumigatus*, *Penicillium citrinum*, and *Fusarium oxysporum*) and the synthesised ZnO NPs also exhibited good antibacterial activity. The synthesised ZnO NPs were crystalline with a hexagonal wurtzite structure, which is the typical structure of ZnO, and the particle size was between 30 and 100 nm. These studies demonstrate that ZnO NPs can be synthesised using microbe-assisted synthesis. However, isolation and growth of the microbes can be time-consuming, and contamination should be strictly avoided during the process; thus, it requires very skilled personnel to handle the process [39]. This has led to research efforts more focused on plant-mediated synthesis as plants are readily available, and their extracts can be extracted with simpler processes.

#### 2.3.1. Plant-Mediated Synthesis

ZnO NPs can be successfully prepared using aqueous plant extracts from leaves, flowers, roots, stems, and barks. This is due to the presence of phytochemicals in plants that are capable of functioning as reducing agents as well as stabilising or capping agents during the fabrication of NPs. These phytochemicals include terpenoids, flavonoids, phenolic compounds, saponins, alkaloids, tannins, carbohydrates etc. They have been reported to be responsible for reducing Zn salt precursors and stabilising ZnO NPs during synthesis. Plant-mediated synthesis is a green synthesis method that is biocompatible, low-cost, environmentally acceptable, and simple and can be easily scaled up for commercial production of NPs [14,15].

Upadhyay et al. [42] conducted a comparison study of structural, morphological, and optical properties of ZnO NPs synthesised using conventional chemical methods and leaf extracts of *Ocimum tenuiflorum*. The study reported that green synthesis using plant extracts resulted in better properties of ZnO NPs as compared to chemical methods. In another study, Gunalan et al. [43] green synthesised ZnO NPs using *Aloe vera* extract. The *Aloe vera* extract synthesised ZnO NPs had higher antibacterial activity than the bulk ZnO and ZnO NPs prepared using the chemical route. These studies demonstrate that ZnO NPs can successfully be prepared using safer, simpler, and cost-effective methods. Additionally, plant-mediated synthesis of ZnO NPs has gained attention in the biomedical field over the years, and this is mainly due to the plant-mediated ZnO NPs exhibiting greater biological activity than plant extracts, bulk ZnO and ZnO NPs prepared via chemical means.

Several studies have proposed mechanisms for the plant-mediated synthesis of NPs, and examples of proposed mechanisms are shown in Figure 3. Generally, the phytochemicals in the plant extract convert the aqueous Zn salt precursors to Zn via a reduction reaction and then oxygen is added during calcination whilst some of the phytochemicals stabilise the synthesised NPs [44]. The second proposed mechanism involves the formation of a complex between Zn^2+^ and the phytochemicals, followed by the formation of Zn(OH)_2_ via hydrolysis and then the formation of ZnO NPs after calcination [45].

The phytochemicals in plant extracts vary in composition depending on the extraction method, plant parts used, and plant used; hence, it is difficult to deduce the mechanism of ZnO NPs formation. Moreover, because plant extracts contain a variety of compounds, it is difficult to determine which compounds are responsible for the reduction and NP stabilisation [12]. As a result, it makes it difficult to deduce the plant-mediated synthesis mechanism of NPs formation. The synthesis method is affected by phytochemical composition, temperature, pH, reaction time, and precursor concentration, which inherently affects the quantity and quality of the synthesised NPs [46].

#### 2.3.2. Plant-Mediated Synthesis of ZnO NPs Procedure

Different morphologies, such as nanowires, nanorods, nanospheres, and nanotubes, can be obtained by controlling the preparation parameters of ZnO NPs [47]. The most common synthesis procedure for plant-mediated ZnO NPs, which is summarised in Figure 4, begins with plant collection and cleaning of the plant parts such as leaves, roots, stems, and flowers with distilled water to remove impurities. This is followed by drying, grinding the plant to powder, and extracting the phytochemicals using solvents, most often distilled water, to obtain the plant extract. The plant extract is then mixed with Zn salt precursors such as zinc nitrate hydrate, zinc chloride, and zinc acetate to obtain a precipitate. The mixing of Zn precursor and the plant extract is carried out at optimal pH, temperature, and reaction time. The precipitate is then calcinated to obtain the ZnO NPs.

#### 2.3.3. Factors Affecting Synthesis of Plant-Mediated ZnO NPs

There are several factors that are known to affect the preparation of plant-mediated ZnO NPs, and these factors subsequently influence the yield, size, morphology, and applications of the synthesised ZnO NPs. These factors include pH, temperature, reaction time, and mixing ratios of Zn precursor and plant extract [12]. Padalia et al. [48] reported the effect of pH on the size and morphology of *Salvadora oleoides*-mediated ZnO NPs. According to the Transmission Electron Microscope (TEM) analysis, the *Salvadora oleoides*-mediated synthesis at pH 5 produced spherical and round ZnO NPs with an average particle size of 26.62 nm. Whereas the TEM analysis revealed that ZnO NPs synthesised at a pH of 8 resulted in irregularly shaped NPs with an average particle size of 38.62 nm.

In another study, Umamaheswari et al. [49] investigated the influence of pH on the synthesis of ZnO NPs using *Raphanus sativus var. Longipinnatus* extract. The effect of pH on the synthesis process was studied from 8 to 14 using Ultraviolet-Visible spectroscopy (UV-Vis). The authors reported that no absorption peaks were observed at lower pH between 8 and 10 and at higher pH of 14. For pH 12, an absorption peak was observed at 369 nm, indicating ZnO NPs formation. These observations highlight the significance of pH in plant-mediated synthesis of ZnO NPs. This is because the phytochemicals that act as reducing agents and stabilising agents can be affected by pH changes, and this will subsequently affect their reducing/stabilisation ability [50]. This affects ZnO NPs formation, which may result in different morphology, size and yield of the ZnO NPs. Thus, the optimal pH is dependent on the phytochemicals present in the plant extract that act as the reducing agents or stabilising agents. However, it has also been reported that the higher the pH, the greater the reducing ability of the reducing agents [51].

Calcination temperature is another synthesis parameter that has an effect on the morphology, size, and properties of ZnO NPs. Sukri and associates [52] studied the influence of calcination temperature on the morphology and size of ZnO NPs synthesised using *Punica granatum* fruit peel. The synthesised ZnO NPs were calcined from 400 to 700 °C, and the authors reported that increasing calcination temperature produced irregularly shaped NPs and an increase in the size of NPs, whereas spherical NPs were observed at lower temperatures using TEM. The general observation from the BET surface analysis was as the calcination temperature increased, the specific surface area decreased. Mohammadi and Ghasemi [53] investigated the effect of reaction temperature on the synthesis of ZnO NPs. The authors synthesised ZnO NPs using cherry leaf extracts at 25, 60, and 90 °C. The particle size of synthesised NPs increased (87.5–116 nm) as temperature increased. The authors also investigated the effect of concentration of the Zn precursor on the synthesis of ZnO NPs. The particle size increased from 20.7 to 96.5 nm as the concentration of the Zn(NO_3_)_2_·6H_2_O precursor (0.005, 0.02, 0.05, and 0.3 M) increased. Different plants contain different phytochemicals; therefore, the reducing and stabilising compounds available for the synthesis of NPs differ from plant to plant [39]. This results in different reducing abilities of plant extracts in reducing Zn salt precursors, thus affecting the synthesis process. Therefore, the effect of synthesis conditions is strongly related to the plants utilised for the synthesis of ZnO NPs via the green synthesis route. Thus, the optimal conditions for the synthesis of ZnO NPs are strongly dependent on the nature of the phytochemicals present in the plant extract. However, Xu et al. [13] state that the general trend as the concentration of Zn salt precursor and plant extract increases results in a decrease in ZnO NPs particle size, while an increase in calcination temperature and reaction temperature results in a decrease in ZnO NPs particle size.

## 3. Green Synthesis of ZnO NPs Using Some Indigenous Plant of Southern Africa

### 3.1. Indigenous Medicinal Plants of Southern Africa

The African continent is rich in fauna and flora. It has an estimated 45,000 plant species, with 5000 of the plant species having medicinal value. Africa experiences subtropical and tropical climates, and this gives rise to different kinds of secondary metabolites being produced by plants to cope with the harsh environment. Moreover, it also receives large amounts of sunlight throughout the year, and this leads to microbes, plants and animals producing more chemopreventive substances, such as secondary metabolites, than other parts of the world [54]. This means that African plants may be a source of unique secondary metabolites. However, the exploration of African plant species for biomedical application and green synthesis of NPs has been limited.

The southern part of Africa is known to be rich in indigenous flora. It is home to an estimated 22,755 plant species with an estimated 3000 plant species with traditional medicinal value, which is about 13.8% of the flora [55]. Aboyewa et al. [56] reviewed some of the indigenous medicinal plants from southern Africa that have been utilised for the preparation of metallic NPs and their biological applications. The review included medicinal plants such as *Lessertia frutescens* (*Sutherlandia frutescens*), *Aspalathus linearis*, *Salvia africana-lutea*, *Galenia africana*, *Catharanthus roseus*, *Hypoxis hemerocallidea*, and *Cotyledon orbiculate*. Considering the vast diversity of indigenous medicinal plants found in southern Africa, only a small portion has been explored for NPs synthesis. The indigenous medicinal plants are still understudied and underutilised.

### 3.2. Plant-Mediated Synthesis of ZnO NPs Using Some Indigenous Medical Plant Extract of Southern Africa and Their Applications

This section will discuss the indigenous medicinal plants from southern Africa that have been explored for the fabrication of ZnO NPs and their various applications. Table 1 summarises the extraction conditions, Zn salt precursors, synthesis conditions, and average particle size of the ZnO NPs synthesised using indigenous medicinal plants from southern Africa, which will be discussed in the sections below.

#### 3.2.1. *Agathosma betulina*

*A. betulina*, also known as Buchu, is a medicinal plant native to South Africa. It is a perennial shrub in the Rutaceae family that grows to heights of 2.0 m. Buchu can be found in the Western Cape province, in the lower elevations of the mountainous areas [57]. The Buchu plant is highly commercialised due to the medicinal benefits of its essential oils and the leaf extracts. The essential oils are used in the cosmetics, pharmaceuticals, food, and beverage industries, while the non-volatile leaf extracts are used for teas, capsules, and herbal water. The non-volatile leaf extracts are used for stomach pains, cough, flu, rheumatism, kidney, and urinary tract infections [58].

The Buchu essential oils and leaf extracts have exhibited biological activity against microbials such as *Bacillus cereus* (*B. cereus*), *Klebsiella pneumoniae* (*K. pneumonia*), *Enterococcus faecalis* (*E. faecalis*), *Candida albicans* (*C. albicans*), and *Staphylococcus aureus* (*S. aureus*) [59]. Despite so many reports on the isolated compounds of the Buchu volatile extracts, the phytochemistry of the non-volatile extracts is still understudied. However, the bioactive phytochemicals such as flavonoids, tannins, monoterpenoids and triterpenoids have been reported, and these can be used for the capping and stabilisation of NPs [60].

Thema et al. [22] utilised *A. betulina* leaf extracts for the green synthesis of ZnO NPs. The authors reported highly crystalline and quasi-spherical agglomerated ZnO NPs calcined at 500 °C, as confirmed by TEM and the Selected Area Electron Diffraction (SAED) profile, respectively. Figure 5 depicts the morphology and microscopy results of the synthesised ZnO NPs. X-ray diffraction (XRD) confirmed a single-phase hexagonal ZnO wurtzite structure with an average particle size of ~15.8 nm.

Fourier Transform Infrared (FTIR) spectroscopy is an important tool for identifying functional groups. FTIR analysis of both plant extracts and synthesised NPs can be used to identify functional groups involved in the reduction and capping/stabilisation of synthesised NPs. The authors did not report on the FTIR of the *A. betulina,* but they did, however, report on the Attenuated-Total Reflection—Fourier Transform Infrared spectroscopy (ATR-FTIR) of the synthesised ZnO NPs. The ATR-FTIR results of the synthesised ZnO NPs revealed bands at 3437.5 (OH stretch), 493.8, and 846.2 cm^−1^, with the former attributed to water adsorbed on the ZnO NPs and the last two attributed to Zn–O stretching mode. There were no bands from functional groups of the phytochemicals in *A. betulina* extracts indicating there were no phytochemicals incorporated into the synthesised ZnO NPs. The synthesised ZnO NPs were investigated for varistor response, and they demonstrated good varistor activity with an electric field of 344.9 and 408.1 V/mm at 0.1 and 1 mA/mm^2^, respectively [22].

Although *A. betulina* was successfully used in the synthesis of ZnO NPs, more research is needed, such as optimisation of the phytochemical extraction process and other synthesis process conditions, as the authors only optimised the ZnO NPs annealing temperature. This is significant because it may affect the size and morphology of the NPs and, thus, their applications. Furthermore, the mechanism of ZnO NP synthesis using *A. betulina* extracts should be investigated. In addition to *A. betulina* being successfully utilised for synthesising ZnO NPs, it has also been explored for the preparation of CdO NPs [60], NiO [61] and CeO_2_ [62], with the synthesised CeO_2_ being used for antibacterial applications.

#### 3.2.2. *Plumbago auriculata*

*P. auriculata* also known as Cape Leadwort or Cape Plumbago is an endemic South African evergreen perennial shrub that is most commonly found in the country’s Western Cape province. Although *P. auriculata* is native to South Africa, it can also be found in warmer climates around the world, such as tropical and subtropical regions. It is a member of the Plumbaginaceae family and is known to possess medicinal benefits. The Plumbago genus consists of 18 species, and *P. auriculata* is the least studied species in the genus [63,64].

Traditionally, *P. auriculata* has been used for the treatment of warts, fractures, malaria, emetics, piles, diarrhoea, skin lesions, oedema, and rheumatism. Additionally, it has been reported to have pharmacological properties such as antimicrobial, anticancer, antiulcer, antimalaria and antifungal [63]. Tannins, flavonoids, phenols, carbohydrates, proteins, alkaloids, and saponins are the phytochemicals that have been identified in methanolic extracts of *P. auriculata* leaves [65]. Tharmaraj and Antonysamy [66] reported the presence of phenolics, tannins, saponins, carbohydrates, steroids, flavonoids, terpenoids in organic extracts of the aerial parts while the aqueous extracts only showed the presence of tannins. These phytochemicals may be utilised in the green synthesis of NPs and can act as capping agents or stabilising agents during synthesis.

Melk et al. [23] reported green synthesised ZnO NPs using alcoholic extracts of *P. auriculata* aerial parts. The synthesised ZnO NPs were observed to be spherical with some agglomerated in shape; they were hexagonal in phase as confirmed by Scanning Electron Microscope (SEM) and XRD, respectively, and a negative Zeta potential with an average particle size of ~38.3 nm. The reaction between the Zn precursor (zinc acetate) and *P. auriculata* was maintained at a pH of 12, which is the pH that has been speculated to be suitable for the plant-mediated synthesis of ZnO NPs [13]. However, since temperature and pH are known to affect the size and morphology of ZnO NPs, it would have been interesting to note how reaction temperature and pH affected the synthesis of the *P. auriculata*-mediated ZnO NPs.

The High-Performance Liquid Chromatography (HPLC) screening of the *P. auriculata* extract identified gallic acid, chlorogenic acid, catechin, and ellagic acid as some of the major compounds and some of the compounds that could be responsible for the reduction and stabilisation of the synthesised ZnO NPs. This is similar to the HPLC screening of aqueous leaf extracts of *Pelargonium odoratissimum* reported by Abdelbaky et al. [67]. The HRTEM particle size of the ZnO NPs synthesised using the *P. odoratissimum* extracts was reported to be 34.12 nm, which is comparable to the reported particle size of the *P. auriculata*-mediated ZnO NPs. The FTIR spectrum revealed bands at 1724, 1619, 1449, and 1371 cm^−1^ for the aerial parts extracts of *P. auriculata,* which were attributed to CH bending vibrations and C–C (in-cycle) or C=O stretch in the phenolic compounds. These frequencies exhibited reduced intensity in the FTIR spectrum of the synthesised ZnO NPs due to the attachment of the phenolic and amino groups. The peak at 561 cm^−1^ was assigned to the *P. auriculata*-mediated ZnO NPs and the UV-Vis spectrum of the *P. auriculata*-mediated ZnO NPs exhibited absorbance at 343.32 nm [23].

The synthesised ZnO were investigated for antiviral activity against avian metapneumovirus (_a_MPV), which causes respiratory problems in humans and animals. The antiviral activity was observed to be 42.64 ± 4.08 µgmL^−1^ for MOI of 0.001 D_50_/cells, which was remarkable. The mechanism of action of ZnO against _a_MPV was attributed to the binding of the Zn to the M2-1 viral protein, which is responsible for the replication and pathogenesis of the virus in vivo [23]. To the best of our knowledge, this is the first report of plant-mediated ZnO NPs being investigated for antiviral activity against _a_MPV. However, further studies are required, e.g., optimisation of the synthesis of ZnO NPs, mechanism of ZnO NP formation, and in vivo trials to fully understand and control the synthesis process of ZnO NPs using *P. auriculata* extracts. *P. auriculata* has also been explored for the synthesis of Ag NPs, and the synthesised Ag NPs were assessed for antimicrobial application [68].

#### 3.2.3. *Monsomia burkeana*

*M. burkeana* is a medicinal plant in the Geraniaceae family and is indigenous to countries in the Southern parts of Africa, including Botswana, Lesotho, Mozambique, South Africa, Eswatini, and Zimbabwe. The plant is popularly known as a special tea in Limpopo province of South Africa, mainly in rural areas, and its decoctions are used for blood cleansing, improving erectile problems, and improving libido (IKS) in males [69,70,71]. This special tea has been reported to contain volatile oils, vitamins, carbohydrates, flavonoids, polyphenols, purines, and alkaloids. Moreover, the special tea has been reported to contain active antibacterial compounds, which can be used for biomedicine applications [72]. Phytochemical screening of M. *burkeana* leaf extracts using Liquid Chromatography-Mass Spectrometry (LC-MS) detected the presence of tannins, flavanol glycoside and phenol constituents, and the possible compounds deduced from the mass to charge ratio were granatin, ellagic acid and quercetin-3O-β-xylopyranosyl-(1,2)O-β-galactopyranoside [73]. The phytochemicals present in the special tea can also be used for the green synthesis of NPs.

Ngoepe and associates [24] successfully synthesised ZnO NPs using extracts of *M. burkeana* and explored their use for photodegradation of methylene blue dye, antibacterial and anticancer activity. The synthesised ZnO NPs were hexagonal in structure, crystalline, and their particle sizes were between 5 and 15 nm, as shown in Figure 6. The effect of the reaction conditions, such as temperature and pH, were not reported. The UV-Vis and FTIR spectra of the synthesised NPs revealed a strong absorbance at 325 nm and a small peak at 500 cm^−1^, respectively. Moreover, the FTIR spectrum of the as-synthesised ZnO NPs also exhibited bands for OH, HO–CO, H_3_CO and CH, which are due to the phytochemicals from the plant extracts, and these can enhance the biological activity of the synthesised ZnO NPs.

The photodegradation of methylene blue dye using the synthesised ZnO NPs only managed to reach 48% degradation after 45 min [24]. This was very low when compared to other reports of photodegradation of methylene blue using ZnO NPs synthesised using medicinal plants. Kahsay [74] reported 71.53% maximum degradation of MB dye at 180 min using *Becium grandiflorum* synthesised ZnO NPs, while Faisal and co-workers [75] reported 88% degradation of MB at 140 min using fruit extracts of *Myristica fragrans* synthesised ZnO NPs, and Khan et al. [76] observed 93.25% degradation at 70 min using *Passiflora foetida* synthesised ZnO NPs. The low photodegradation efficiency observed for the *M. burkeana* synthesised can be enhanced by doping the ZnO NPs with metals, such Fe, Al, and Cu, or fabricating the composite of ZnO with other metal oxides.

Despite the *M. burkeana* synthesised ZnO NPs showing poor photocatalytic activity, the NPs exhibited good antibacterial activity. The synthesised ZnO NPs were active against *Escherichia coli* (*E. coli*), *Pseudomonas aeruginosa* (*P. aeruginosa*) and *S. aureus*. However, the synthesised ZnO NPs exhibited no antibacterial activity against *E. faecalis*. Moreover, the synthesised NPs exhibited antiproliferative properties against A549 lung cancer [24]. Therefore, *M. burkeana* extracts can successfully be used for synthesising ZnO NPs, which can then be used for anticancer applications. However, there is a need for optimisation of the synthesis of the *M. burkeana*-mediated ZnO NPs and in vivo studies. *M. burkeana* plant extracts have also been used to synthesise NiO NPs [77] and TiO_2_ NPs [73] and explored for photodegradation of organic dyes and antibacterial application.

#### 3.2.4. *Lessertia montana*

*L. montana,* also known as Mountain Balloon Pea and formerly known as *Sutherlandia montana,* is a plant species with medicinal value that is native to South Africa and belonging to the Fabaceae family. It is a soft-wooded shrub with silvery green leaves, large red flowers, bladdery pods and grows to heights of about 0.5–1.0 m. In South Africa, the Basotho people from the eastern Free State province traditionally use the leaf infusion as a sedative and for treatment of heart diseases [78]. Ashafa and Alimi [79] screened *L. montana* leaf decoctions for phytochemicals and reported the presence of saponins, cardiac glycosides, alkaloids, phenols, triterpenes, phytosterols, and flavonoids. These phytochemicals can be utilised as reducing and stabilising agents in NPs synthesis.

The aqueous extracts of *L. montana* leaves have been utilised for the synthesis of ZnO NPs. The synthesised ZnO NPs were assessed for antioxidant and antihyperglycaemic properties [25]. The morphology of the synthesised ZnO NPs was both cubic and spherical, with an average particle size of 13.8 nm. The FTIR spectrum of the *L. montana*-mediated ZnO NPs showed O–H stretching at 3365 cm^−1^, medium C–H stretching at 2945 cm^−1^, strong C–C stretching at 1412 cm^−1^, strong CO–O–CO stretching at 1046 cm^−1^, and N–H stretching at 1596 cm^−1^ bending peaks due to hydroxyl group, alkanes, aromatic, anhydride and amine groups, respectively. The authors attributed that these groups could be from phytochemicals such as alkaloids, phenols and flavonoids, which have amine and aromatic alcohols, respectively. These phytochemicals could have been involved in the stabilisation of the synthesised NPs [25]. However, the authors did not report the mechanism of formation of the ZnO NPs using the *L. montana* extract. Further studies can also be carried out to identify the compounds in the *L. montana* extract responsible for the functional groups reported from the FTIR spectrum.

The synthesised ZnO NPs exhibited great antioxidant properties as shown by the half maximal inhibitory concentration of 120.31 µg/mL for 2,2-diphenyl-1-picrylhydrazyl (DPPH), 711.45 µg/mL for 2,2′-Azino-bis(3-Ethylbenzothiazoline-6-Sulfonic Acid) (ATBS) and 184.16 µg/mL for metal chelating when compared to bulk ZnO, and *L. montana* leaf extracts. However, the quercetin control performed better than the synthesised ZnO NPs for ATBS and metal chelating. The synthesised ZnO NPs exhibited strong antihyperglycaemic activity with IC_50_ values of 0.0037 gL^−1^ for α-glucosidase, while α-amylase had the mildest inhibition with 0.620 gL^−1^ inhibition. The authors reported that the mechanism of action of synthesised ZnO NPs on the α-amylase was competitive while that for α-glucosidase was non-competitive [25].

#### 3.2.5. *Lessertia frutescens*

*L. frutescens,* commonly known as cancer bush and formerly known as *Sutherlandia frutescens* is a medicinal plant that belongs to the Fabaceae family. It is indigenous to southern Africa, in South Africa, Botswana, Namibia, and Lesotho. *L. frutescens* is a perennial shrub that grows to heights of about 0.3–3 m with orange to red flowers, silvery green-grey pinnate leaves, and bladder-like fruit pods [80,81].

The stems, roots, leaves, and flowers of *L. frutescens* have been widely used as traditional medicine as infusions or decoctions for treatment of cancer, fever, wounds, stress, depression, eczema, urinary tract infections, influenza, eye infections, rheumatism, back pain, etc. In recent years *L. frutescens* has found use in boosting immunity in people living with HIV/AIDS and improving their way of life [81,82]. Unlike *L. montana* mentioned earlier, *L. frutescens* is a widely studied traditional medicinal plant.

The pharmacology studies of *L. frutescens* have validated traditional use claims with pharmacologic properties, including antiviral, antibacterial, antioxidant, anti-inflammatory, antithrombotic, antistress, antidiabetic, antiproliferative, and anticancer. Moreover, phytochemical compound screening of *L. frutescens* has identified flavonol glycosides, triterpenoids, cycloartane glycosides, flavonoids, canavanine, pinitol, and γ-aminobutyric acid (GABA), which could be responsible for the pharmacologic properties of *L. frutescens* [83]. These phytochemicals can also be used in the synthesis of NPs for various applications.

The phytochemicals in *L. frutescens* were utilised in the synthesis of ZnO NPs, and the resulting ZnO NPs were investigated for antibacterial and anticancer application [26]. The synthesised ZnO NPs were spherical with some agglomeration and had an average particle size of 13.3 nm. The FTIR spectrum exhibited a band at 1400 cm^−1^, which was due to ZnO NPs. There was still the presence of secondary metabolites on the synthesised ZnO NPs even after calcination at 700 °C, which is evident from the FTIR spectrum. The FTIR spectrum of the synthesised ZnO NPs revealed peaks similar to those found in the plant extract spectrum, such as C=N, O–H, and N–H functional groups, indicating that the phytochemicals were incorporated into the synthesised ZnO NPs. GABA, flavanol, canavanine, sutherlandian B, and sutherlandioside B were the phytochemicals present in the *L. frutescens* extract identified using LC-MS. The FTIR spectrum of *L. frutescens* extract revealed the presence of OH, NH_2_, H_3_CO, and CH functional groups, confirming the presence of these polyphenols. The authors proposed that the polyphenols were responsible for the reduction in the Zn salt precursor [26].

The antibacterial activity was assessed against *E. coli*, *S. aureus*, *P. aeruginosa*, and *E. faecalis.* The synthesised ZnO NPs managed to inhibit all the strains; however, they exhibited remarkable antibacterial activity against Gram-negative bacteria with 95% inhibition against *E. coli* and 100% inhibition against *P. aureginosa*. Additionally, the synthesised ZnO NPs exhibited strong anticancer activity against human lung cancer A549 with 93.4% maximum cell death at 1000 µg mL^−1^ [26]. The strong biological activity of the synthesised ZnO NPs can be attributed to the small size of the NPs, which translates to a large surface area and the secondary metabolites incorporated in the NPs. *L. frutescens* has also been investigated for the synthesis of NiO NPs [84], AgO NPs [85], ZnS NPs [86], CdS NPs [87], ZnO/SnO_2_ NPs [88] and Ag NPs [89] for various applications such as photodegradation of organic dyes and pharmaceutical pollutants, antibacterial activity, and anticancer activity.

#### 3.2.6. *Tulbaghia violacea*

*T. violacea* is a plant species that has medicinal benefits belonging to the Alliaceae family, also known as “wild garlic or society garlic”, that is native to southern parts of Africa, including South Africa, Zimbabwe, Lesotho, Botswana, and Mozambique. It is a bulbous plant that is characterised by hairless leaves arising from a white fleshy stalk, purple flowers, and a distinctive odour similar to *Allium sativum* when the leaves or rhizome are crushed. The plant has been used traditionally to treat ailments such as headaches, fever, constipation, rheumatism, heart problems, chest problems, fits, and paralysis [90,91,92,93]. Madike et al. [94] reported on the phytochemical screening of the aqueous leaf and bulb extracts of *T. violacea* and showed the presence of saponins, cardiac glycosides, coumarins, terpenoids, proteins, flavonoids, and phenols. These secondary metabolites can be utilised for the synthesis of NPS for various applications.

Mbenga and co-workers [27] successfully synthesised ZnO NPs using *T. violacea* bulb extract and investigated the cytotoxicity of the synthesised NPs. The synthesised ZnO NPs were highly crystalline with hexagonal wurtzite phase, spherical, and closely packed with an average particle size of about 45.26 nm.

The FTIR spectrum of the *T. violacea* bulb extracts exhibited some peaks, e.g., S-H thiol group at 2310 cm^−1^, O-H bending stretch at 1603 cm^−1^, S=O at 1320 cm^−1^ and 1015 cm^−1^, which were attributed to thiols and cysteine amino acid present in the extract. However, the authors did not report on the FTIR of the *T. violacea*-mediated ZnO NPs. It would have been interesting to note the functional groups that were possibly incorporated into the ZnO NPs since they are known to improve biological properties of the NPs. However, the thiols present in *T. violacea* bulb extract could be responsible for the reduction in the Zn precursor (Zn(CH3COO)2·2H2O) since thiols are known to be nucleophilic and, hence, have antioxidant properties [95]. The cysteine amino acid could have been responsible for the stabilisation of the ZnO NPs since amino acids contribute to the stabilisation of growth synthesis NPs [96]. However, further studies are required to establish the mechanism of formation of the ZnO NPs. Additionally, there is a need to optimise the synthesis method as reaction parameters have an effect on the size and morphology of the NPs. It would be interesting to note how it will affect the cytotoxicity of the synthesised ZnO NPs. This is because the dissolution of Zn depends on the particle size of the ZnO NPs. The larger the particle size, the higher the cytotoxicity [97]. Mbenga et al. [27] reported a slight reduction in the toxicity of ZnO NPs when exposed to human liver cells and was noted as evident from the IC_50_ of 6.47 × 10^−1^ µgmL^−1^ for *T. violacea*-mediated ZnO NPs compared to pristine ZnO NPs with 4.04 × 10^−1^ µgmL^−1^. To the best of our knowledge, *T. violacea* extracts have not been further explored for the synthesis of other NPs.

#### 3.2.7. *Aspalathus linearis*

*A. linearis* is a flowering shrub that belongs to the Fabaceae family and is commonly known as Rooibos. The plant is native to the Western Cape province of South Africa, and the Khoi-San communities have used the leaves as herbal tea. Currently, Rooibos tea is a commercialised herbal tea that has global success. *A. linearis* has been used traditionally to treat different ailments, including skin ailments, kidney ailments, diarrhoea, hypertension, and colic, and to stimulate appetite [98,99]. *A. linearis* leaves contain two unique phenolic compounds called aspalathin and aspalalinin, as well as other phenolic compounds, including flavones, flavonols and flavanones, as listed in Table 2 [100]. These phenolic compounds can be used as reducing agents responsible for reducing metal ions in the synthesis of NPs.

Diallo et a. [28] successfully prepared ZnO NPs using different aqueous concentrations of *A. linearis* leaf extracts at 80 °C. The authors optimised the concentration of the *A. linearis* extract that was utilised for the synthesis of ZnO NPs with a threshold concentration of 6 g of *A. linearis* per 300 mL of deionised water. However, they did not report on the optimisation of pH and reaction temperature, which are also important as they affect the size and morphology of the synthesised NPs. The synthesised ZnO NPs were quasi-spherical in shape without any agglomeration and amorphous before calcination, as observed in the TEM images SAED pattern, respectively, shown in Figure 7. Highly pure and crystalline ZnO NPs were observed from the XRD analysis after calcination at 350 °C.

Raman spectroscopy studies revealed a hexagonal structure with P6_3_mc symmetry of ZnO with five major peaks. The authors attributed the first three peaks to second-order Raman scattering due to the vibrational modes (3E_2H_–E_2L_ at ~326.1 cm^−1^, A_1(TO)_ at ~379.6 cm^−1^, and E_2H_ at ~431.8 cm^−1^); E_1(LO)_ at ~574.2 cm^−1^ was attributed to a defect due to oxygen vacancies and zinc interstitials, and the peak at ~829.4 cm^−1^ could not be assigned as it was within the background. The authors did not report on the FTIR of the synthesised ZnO NPs, which can be used to identify functional groups responsible for capping the ZnO NPs. The phenolic compounds listed in Table 2 are phytochemicals found in the *A. linearis,* which may be responsible for the reduction in the Zn precursor (Zn(NO_3_)_2_·6H_2_O) since phenolic compounds are known to exhibit great antioxidant activity [96]. However, further studies are required for the identification of compounds present in the *A. linearis* extract using HPLC or LC-MS and studying the mechanism of formation of ZnO NPs. Additional studies are also needed on the optimisation of the synthesis procedure. There were no reports on the use of the synthesised ZnO NPs for any applications; therefore, further studies can also include the application of synthesised ZnO NPs. In addition to being used for the synthesis of ZnO NPs, *A. linearis* has been documented in the synthesis of various NPs, including Co_3_O_4_ NPs [96], Au NPs [98], Eu_2_O_3_ [100], Ag NPs [101] Pd NPs, PdO NPs [102], Rh NPs [103], RuO_2_ NPs [104], SnO_2_ NPs [105], and NiO NPs, [106]. The synthesised NPs were assessed for applications in photodegradation of dyes, water splitting and biological application.

#### 3.2.8. *Dovyalis caffra*

*D. caffra* is a plant species that is indigenous to southern African countries such as South Africa, Malawi, Zimbabwe, Mozambique, Botswana, and Eswatini. It is a member of the Salicaceae family and is commonly known as Kei apples. In South Africa, *D. caffra* is abundant in the Eastern Cape province, particularly along the Kei River. *D. caffra* is best known for its aromatic apricot-like fruits, which can be consumed raw or processed into jams, alcoholic beverages, and jellies [107]. The bark, thorns, and roots of *D. caffra* are used as traditional medicine in southern Africa, while the fruits are used as traditional medicine in Kenya [108]. Qanash et al. [109] investigated phytochemicals present in the phenolic and flavonoid contents of *D. caffra* fruit methanolic extracts using HPLC. The authors reported the presence of chlorogenic acid as the major constituent, catechin as the second major constituent and gallic acid as the third major constituent of the phenolic and flavonoid contents of *D. caffra* fruit extracts. Compounds such as Luteolin, Apegenin, Aberiamide, and β-Amyrin are flavonoids that have been identified in the leaf extracts *of D. caffra* [108]. All these compounds can be utilised in the preparation of ZnO NPs.

The leaves of *D. caffra* have been utilised for the synthesis of ZnO NPs. The *D. caffra* leaf-mediated ZnO NPs had a hexagonal wurtzite structure and particle size of 25.29 nm from the XRD analysis. The SEM analysis resulted in densely packed hexagonal rod-like NPs, while the TEM analysis obtained rod-like and spherical ZnO NPs [29]. The authors proposed the same mechanism of ZnO NPs formation as the one reported by Adeyemi et al. [30]. Studies of the effect of synthesis conditions on the size and morphology of the synthesised ZnO NPs were not reported. The synthesised ZnO NPs were investigated for antioxidant activity using the DPPH method and cytotoxicity activity against the MCF7 breast cancer cell line. The *D. caffra* leaf-mediated ZnO NPs exhibited lower antioxidant activity (IC_50_ = 8.99 mg/mL) as compared to ascorbic acid control (IC_50_ = 4.96 mg/mL) and slightly lower cytotoxicity activity (IC_50_ = 3.90 µg/mL) as compared to the standard drug (IC_50_ = 3.71 µg/mL) [29]. The antioxidant activity can be improved by reducing the size of the synthesised ZnO NPs, which can be achieved by changing the synthesis parameters. This is because the antioxidant activity can be influenced by the size and, consequently, the specific surface of the NPs [110]. The cytotoxicity activity can be improved by doping or compositing the ZnO with other metal oxides. CuO-ZnO NPs synthesised from *D. caffra* leaf extracts were reported to exhibit better antioxidant and cytotoxicity than ZnO NPs [29].

Adeyemi and associates [30] investigated the use of *D. caffra* fruit extracts for the synthesis of ZnO NPs and utilised the synthesised ZnO NPs for the photodegradation of MB dye. The *D. caffra* fruit-mediated ZnO NPs were crystalline and had a hexagonal wurtzite structure, as observed from the XRD analysis. The morphology was rod-like with some agglomeration, and the average length and width were 10.4 and 34.1 nm, respectively, from the TEM analysis. The authors reported a distinct absorption peak at 342 nm, which was the characteristic peak of the *D. caffra* fruit-mediated ZnO NPs from the UV-Vis spectrum. There were no reports on the FTIR analysis of *D.caffra* fruit-mediated ZnO NPs, which could have shown any functional groups from the *D. caffra* fruit extract present on the ZnO NPs. However, the authors reported a possible mechanism of formation of the ZnO NPs whereby the Zn^2+^ from the Zn salt precursor form a zinc complex due to a hydroxyl group from the aromatic compounds in the *D. caffra* fruit extract via hydrolysis. They proposed that phytochemicals such as catechin and gallic acid found in the *D. caffra* fruit extract could have been responsible for the formation of the Zn(OH)_2_, which was then calcined to obtain ZnO NPs, while other phytochemicals present in the extract could have been responsible for the stabilisation of the synthesised ZnO NPs [30].

There is a need for further studies, such as studies on the effect of pH, calcination temperature, and reaction temperature, which are known to affect the size and morphology of synthesised NPs. The synthesised ZnO NPs were investigated for photodegradation of MB dye, and the maximum degradation was reported to be 71% after 120 min [30]. Even though the *D.caffra* fruit-mediated ZnO NPs managed to successfully degrade the MB dye, the authors did not report on the effect of pH, contact time, dosage of ZnO NPs, etc., which are known to affect the photodegradation process.

#### 3.2.9. *Athrixia phylicoides* DC

A. *phylicoides* is a medicinal plant belonging to the Asteraceae family that is native to South Africa. It is commonly known as bush tea and has been traditionally used by the BaSotho, Zulu, Venda and Xhosa people to treat ailments such as sores, hypertension, diarrhoea, boils, acne, and diabetes. Commercialisation of *A. phylicoides* has been reported, but unlike *A. linearis,* it is still in its early stages [111,112]. Some of the compounds that have been isolated and identified in various extracts of *A. phylicoides* leaves include quercetin, chlorogenic acid, 6-hydroxyluteolin-7-O-β-glucoside, and oleanolic acid. These compounds could be utilised for the preparation of NPs for various applications.

Kaningini et al. [31] studied the effects of Zn salt precursor concentration, calcination temperature, and doping on the optical properties of *A. phylicoides* leaf extract-derived ZnO NPs. The Energy Dispersive X-ray Spectroscopy (EDS) results revealed that when dopant concentration was increased, it resulted in Ag and Ce not being fully incorporated into the ZnO NPs. The general trend in the Field Emission Scanning Electron Microscope (FE-SEM) was that as the Zn salt precursor concentration increased, agglomeration also increased while keeping the spherical morphology. However, at a higher Zn salt precursor concentration, the synthesised ZnO NPs were spherical with a mixture of hexagonal and cubic. At higher dopant concentrations, the FE-SEM analysis of Ce-doped ZnO NPs revealed monodispersed rod-shaped NPs, whereas at lower dopant concentrations, the NPs were highly monodispersed and spherical. The FE-SEM analysis of Ag-doped ZnO NPs revealed quasi-spherical NPs and increasing the concentration of dopant resulted in rod-shaped NPs. The UV-Vis analysis of the synthesised NPs revealed that absorbance increased as the Zn salt precursor concentration decreased [31]. This is in contrast to the observations for *Berberis aristata*-mediated ZnO NPs, where the absorbance increased with increasing Zn salt precursor concentration [113]. However, the authors attributed the increase in absorbance with decreasing Zn salt precursor concentration to a decrease in particle size as the Zn salt precursor concentration decreased. An increase in Ce-dopant concentration resulted in a shift to lower wavelengths, whereas Ag-doping had no effect on the band gap [31]. The authors did not report any applications of the synthesised NPs.

## 4. Conclusions and Future Perspectives

The southern African native medicinal plants that were used to make ZnO NPs and were investigated for diverse uses have been reviewed. The majority of ZnO NPs created from medicinal plants have been investigated for biological uses such as antibacterial and anticancer properties. This is not shocking considering that NPs produced by utilising plant-mediated synthesis were found to have superior biological characteristics in vitro compared to NPs made using traditional chemical techniques and plant extracts. In addition to biological applications, the varistor application and photodegradation of organic dyes have been investigated for the ZnO NPs synthesised utilising local medicinal plants. This demonstrates the versatility of ZnO NPs produced by plants.

Southern Africa has a large and diverse flora, with 13.8% having medicinal benefits. However, very few medicinal plants have been explored for their green synthesis of nanoparticles as well as biological applications. This is due to indigenous medicinal plants from southern Africa being understudied and underutilised. The same way goes for ZnO NPs. Very few indigenous medicinal plants from southern Africa have been explored for green synthesis of ZnO NPs. There is a need to explore more indigenous medicinal plants for the synthesis of ZnO NPs and explore the synthesised ZnO NPs for various applications. There is also a need to carry out further studies to optimise the synthesis process of ZnO NPs using indigenous medicinal plants and investigate the mechanism of formation of the ZnO NPs in order to better tailor the NPs for various applications.

## Figures and Tables

**Figure 1 nanomaterials-12-03456-f001:**
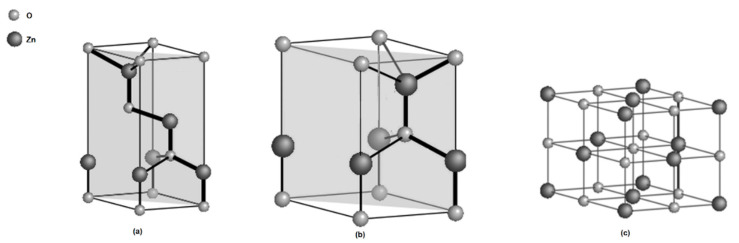
Crystal structure models of ZnO: (**a**) zinc blende, (**b**) wurtzite and (**c**) rock salt.

**Figure 2 nanomaterials-12-03456-f002:**
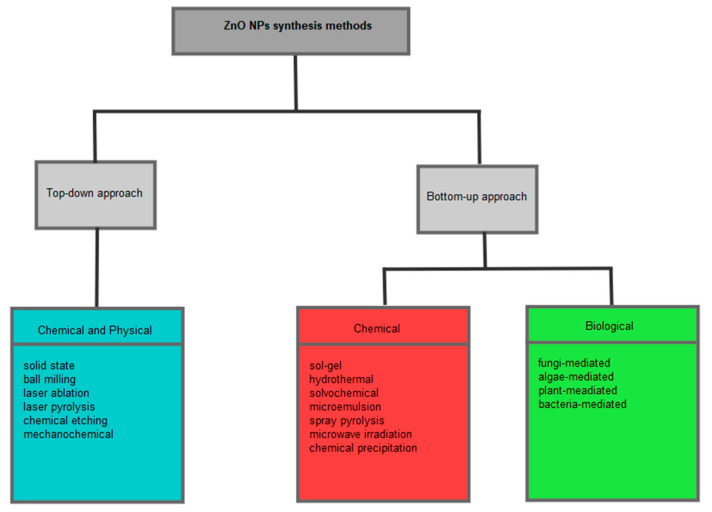
Overview of some of the methods used for ZnO NPs preparation.

**Figure 3 nanomaterials-12-03456-f003:**
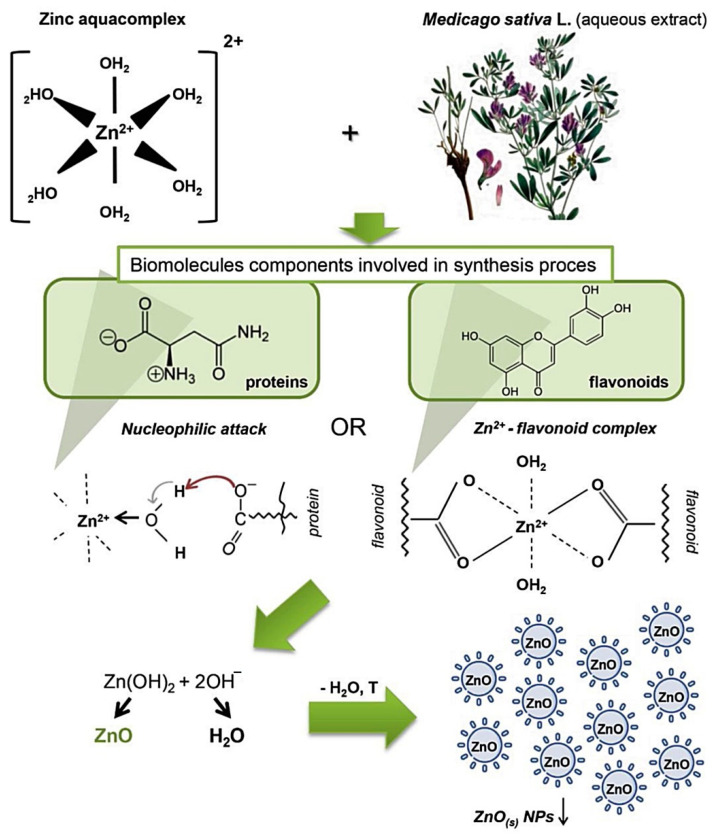
Schematic of the proposed synthesis of ZnO NP using *Medicago sativa* aqueous extracts. Adapted from [45].

**Figure 4 nanomaterials-12-03456-f004:**
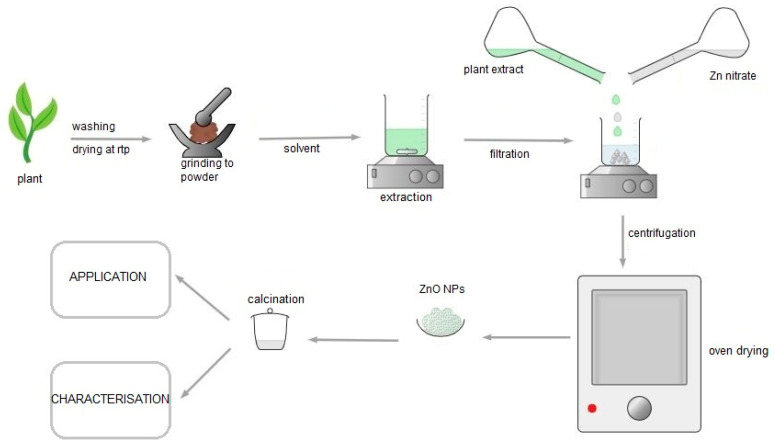
Schematic of the most common ZnO NPs synthesis procedure.

**Figure 5 nanomaterials-12-03456-f005:**
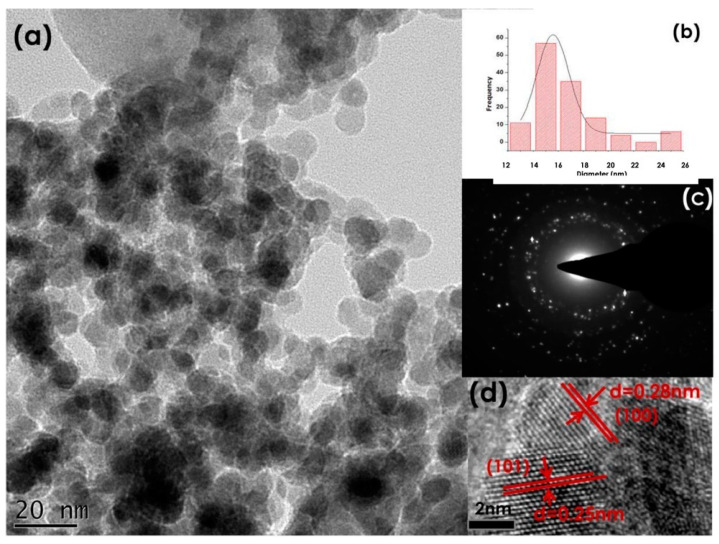
(**a**) TEM image of the synthesised ZnO NPs (**b**) with their size distribution (**c**), their SAED pattern and (**d**) High-Resolution Transform Electron Microscope (HRTEM) image. Adapted from [22].

**Figure 6 nanomaterials-12-03456-f006:**
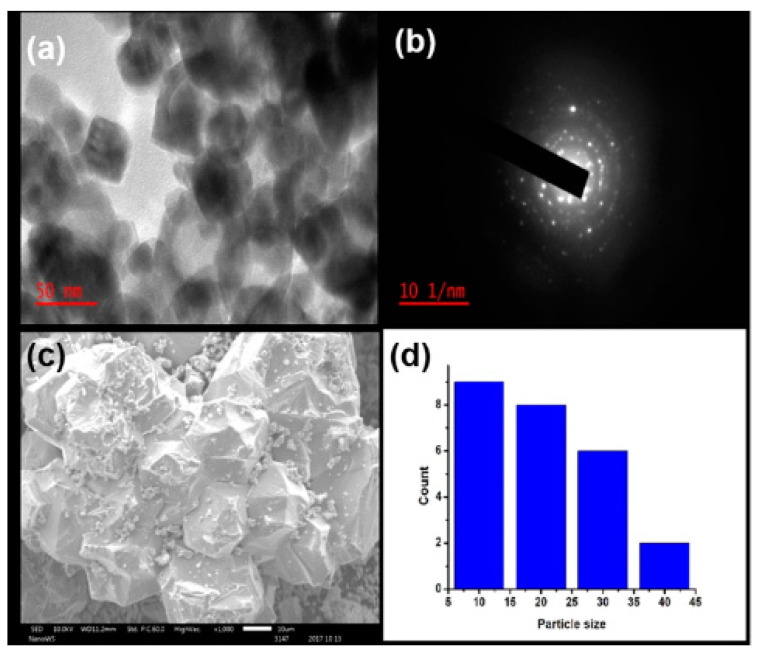
(**a**) TEM image of *M. burkeana*-mediated ZnO NPs, (**b**) with their SAED, (**c**) SEM and (**d**) particle bar graph size. Adapted from [24].

**Figure 7 nanomaterials-12-03456-f007:**
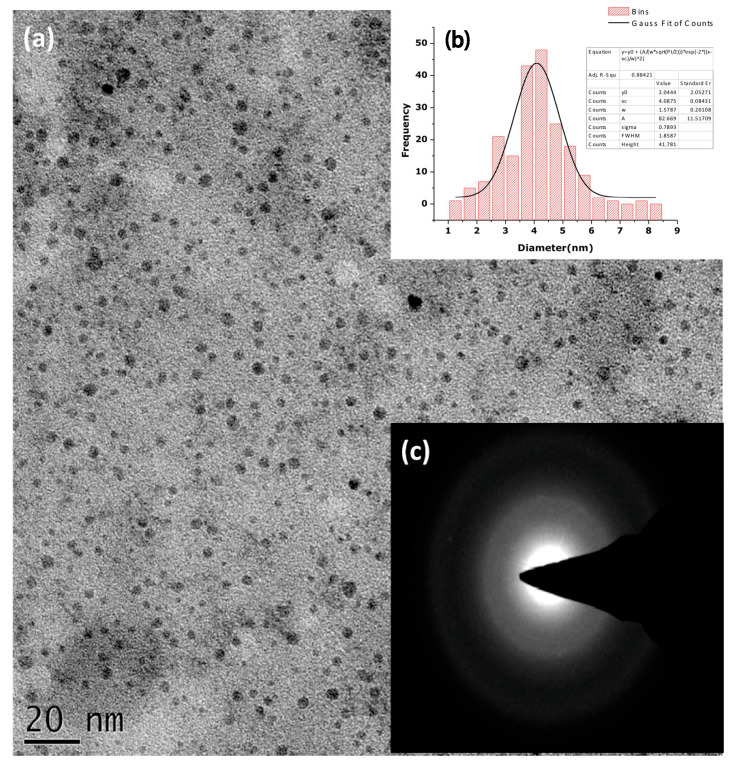
(**a**) HRTEM of the synthesised ZnO NPs (**b**) with their size distribution (**c**) and their SAED pattern. Adapted from [28].

**Table 1 nanomaterials-12-03456-t001:** List of extraction conditions, Zn salt precursors, synthesis conditions, and average particle sizes ZnO NPs synthesised using some southern African indigenous medicinal plants.

Plant and Plant Part	Extraction of Phytochemicals	Zn Salt Precursor	Synthesis Conditions	Average Particle Size	Ref.
*Agathosma betulina* Leaf	Deionised water as solvent, ~100 °C for 1 h, pH 5	Zn(NO_3_)_2_·6H_2_O	100 °C for 2 h, dried at 100 °C, calcined from 100 to 500 °C with 500 °C as optimal	15.8 nm	[22]
*Plumbago auriculata* Aerial parts	Ethanol as solvent, evaporation under reduced pressure using Buchi rotary evaporator	Zn(CH3COO)_2_·2H_2_O	Heated in boiling water bath for 20 min, pH 12, freeze drying	38.3 nm	[23]
*Monsomia burkeana* plant	Deionised water as solvent, 80 °C for 1 h	ZnCl_2_·6H_2_O	80 °C for 1 h, dried at 100 °C, calcined at 700 °C for 1 h	5–15 nm	[24]
*Lessertia montana* Leaf	Distilled water as solvent, 65 °C for 4 h	ZnO	70 °C for 4 h, dried at 50 °C, −80 °C until characterisation	13.8 nm	[25]
*Lessertia frutescens* Leaf	Deionised water as solvent, 80 °C for 15 min	Zn(NO)_3_·6H_2_O	Boiled for 1 h, dried at 80 °C overnight, calcined at 700 °C	13.3 nm	[26]
*Tulbaghia violacea* Bulb	Distilled water as solvent, 80 °C for 1 h	Zn(CH_3_COO)_2_·2H_2_O	80 °C until precipitate formed, pH 12, dried at 50 °C for 3 h, calcined at 350 °C	45.26 nm	[27]
*Aspalathus linearis* Leaf	Deionised water as solvent, 25 °C for 48 h	Zn(NO_3_)_2_·6H_2_O	Room temperature, dried at 80 °C for 2 h, calcined at 300 °C	12.5 nm	[28]
*Dovyalis caffra* Leaf	Deionised water as solvent, 80 °C for 2 h	Zn(CH_3_CO_2_)_2_·2H₂O	85 °C for 1 h, pH 10, dried at 50 °C, calcined at 400 °C for 2 h	25.29 nm	[29]
*Dovyalis caffra* Fruit	Distilled water as solvent, boiled for 20 min	Zn(CH_3_CO_2_)_2_	85 °C for 1 h, calcined at 300 °C for 2 h	34.1 nm * 10.4 nm **	[30]
*Athrixia phylicoides* DC Leaf	Deionised water as solvent, 60 °C until water turned dark green in colour	Zn(NO_3_)_2_·6H_2_O	~80 °C until dark paste formed, calcined at 600 °C and 800 °C	24.5 nm	[31]

* Average length; ** Average width.

**Table 2 nanomaterials-12-03456-t002:** List of some secondary metabolites found in *A. linearis* extracts.

Flavonoid Subgroup	Compound
Flavones	orientin, isoorientin, vitexin, isovitexin, luteolin, and chrysoeriol
Flavanones	dihydro-orientin, dihydro-isoorientin, and hemiphlorin
Flavonols	quercetin, hyperoside, isoquerci-trin, and rutin
Dihydrochalcone	Aspalathin
Cyclic dihydrochalcone	Aspalalinin

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
