# Peer review of "A Review of the Green Synthesis of ZnO Nanoparticles Utilising Southern African Indigenous Medicinal Plants"

_nanomaterials, 2022, doi:10.3390/nano12193456_

Round 1
Reviewer 1 Report
The manuscript summarizes the preparation of ZnO nanoparticles using various African indigenous medicinal plants. This green synthetic approach could result in the formation of environmentally friendly nanoscale materials for diverse potential applications which could provide interesting aspects to the materials science community. Including some additional information in this review could greatly improve the quality of the manuscript.
The authors focused on 7 types of African indigenous plants. As each section (from Section 3.2.1 to Section 3.2.7) describes one type of plant, it is highly recommened to include some Figures to show the shape and distribution of resulting ZnO nanoparticles (including representative images would strength the manuscript).
From each plant extract, including some descriptions on a key ingredient, which mainly influences the formation/stabilization of ZnO nanoparticles, could provide clearer information for readers. For example, the authors stated that “the betulina plant extract possesses bioactive phytochemicals such as flavonoids, tannins, monoterpenoids and triterpenoids and these can be used for capping and stabilisation of NPs” Using this extract resulted in the formation of quasi-spherical agglomerated ZnO nanoparticles. It would be very useful to report which key compound from the extract mainly influenced such formation, if known. This piece of information could provide a very interesting aspect to the materials science community, which could also be utilized in the preparation of other metal oxides.
It is a bit difficult to see the details of Figures (e.g., Figure 3 and Figure 4). Quality should be improved.
Some statement requires references (e.g., The authors stated “Hefny et al. successfully synthesized ZnO NPs using five fun- 135 gal cultures…”, but did not provide any reference)
Author Response
Good day,
Thank you for the review and your contribution to the review paper. As you advised:
- Some figures have been added to show the shape and distribution of resulting ZnO nanoparticles.
- The key compounds from the extract that mainly influenced such formation of ZnO NPs were known.
- Figure 3 was deleted because we could not find a high quality image and quality of Figure 4 has been improved.
- A reference for "Hefny et al. successfully synthesized.." was added.
Kind regards,
Reviewer 2 Report
The emergence of ZnO NPs synthesis by biological methods has meant a significant development in the field of nanoparticles. The interest for this type of synthesis increased rapidly because these methods did not require the use of hazardous/toxic chemicals, as previously employed methods. The use of plants, easily accessible factors, has led to the production of NPs through a simple, fast, cost-effective and eco-friendly process. Synthesis of nanoparticles was attributed to the abundance of biomolecules in plant extracts.
The paper is entitled - A Review of The Green Synthesis of ZnO Nanoparticles Utilizing Southern African Indigenous Medicinal Plants - so it is expected to present, first of all, everything that can be discussed related to the synthesis of nanoparticles and secondly the uses of these nanoparticles.
It is known that the size, morphology, stability and biological properties of NPs are strongly influenced by biomolecules from plant extracts and by experimental conditions, so designing the synthesis of NPs having the desired characteristics is a major area of interest. So, in order to a better understanding of the synthesis, it may be appropriate to describe and discuss the conditions in which these nanoparticles are obtained, the influence of each factor (temperature, pH, reaction time, etc.) on the synthesis and how these factors influence the size of the obtained NPs; with examples for each case.
Also, maybe a subchapter related to methods of analysis used to confirm ZnO NPs synthesis would be useful.
For the second part, of uses, I think that a restructuring of the information is necessary, on each individual utilisation and explained the mechanism through which it exercises its actions.
Other observations:
- - the terms used must be uniform: for example - name of plants must be write in italic style, the same term used ZnO NP or ZnO NPs.
- - tables 1 and 2 repeat certain aspects of the text, from paragraphs 3.2.1-3.2.6 and 3.2.7, so the same information is presented twice, and I don't think it is appropiate.
- - the references in Reference section have been corrected according to Instructions to Authors
- - figures 3 and 4 must be more clear
Please read the paper carefully for English language style and accuracy and make the appropriate corrections and changes.
Author Response
Good day,
Thank you for the review and your contribution to the review paper. As you advised:
- A section was added for the influence of synthesis parameters (temperature, pH, reaction time, etc.) and how these factors influence the size of the NPs with examples.
-
The all used terms were made uniform: for example - name of all plants were italicised, the same term ZnO NPs was used.
-
Table 1 was updated however, we could not change Table 2 as it has some compounds which are not included in the main text.
- The references have been edited according MDPI referencing.
- Figure 3 was deleted because we could find a high resolution image and resolution of Figure 4 has been improved.
-
We could not add indepth mechanism of applications of ZnO NPs as we are currently writing another review paper that focuses on that.
- The paper was carefully read and corrections and changes made were appropriate.
Reviewer 3 Report
The paper is interesting from the point of view on green synthesis of ZnO using plants that originate in South Africa. However, the authors presented only seven papers (and even plants) on the subject. I did the article search and in ten minutes found four more papers on the subject. I recommend that authors do a wider search on the topic and perhaps do not limit their search only on medicinal plants, as there are not so many articles published. Please include critical examination and evaluation of all works and not simple summary of works.
Author Response
Good day,
Thank you for the review and your contribution to the review paper. As you advised:
- Critical evaluation has now been included and not only the summary of the papers.
- We do agree that there is limited exploration of synthesis of ZnO NPs using medicinal plants. However, the purpose of the review has been to bring attention to medicinal plants indigenous to southern Africa by highlighting the work that has been done and what more can be done towards utilising these understudied plants. We are also currently working on another review paper focusing on other plants that have been utilised for ZnO NPs.
Round 2
Reviewer 2 Report
I found in the text most of the changes requested by me and which could be made by the authors. In this format, it is an improved version of the paper and I think it can be published, after the agreement of the editor in charge.
Author Response
Dear Reviewer,
We appreciate the time and effort you put into providing feedback on our manuscript, and we appreciate the insightful comments and valuable improvements to our paper.
Kind regards,
Authors
Reviewer 3 Report
The manuscript was improved, but I believe in wrong direction. My comment was not taken to account. I thought it is a review paper on use of medicinal plants indigenous to Southern Africa for ZnO synthesis, but now, the paper is too explanatory on the use of general plants (not so limited to S.African). I mentioned in my first review that authors should check and add other papers where ZnO was synthetised using native S. African (medicinal) plants. I do not understand why authors did not do this.
I suggest that authors change the title or add more literature on the current title. Here are some examples of papers that authors did not mention in the paper (and I believe there are more, but this would have to be searched by authors):
- https://doi.org/10.1016/j.reffit.2017.05.001 (Passiflora caerulea; native (indigenous) to S.Africa)
- https://doi.org/10.1007/s10904-022-02248-6 (Tulbaghia violacea; netive to S.A.)
- https://doi.org/10.3390/molecules27103206 (Dovyalis caffra; native to S.A.)
- https://doi.org/10.1088/2053-1591/ab1afa (Sutherlandia frutescens ; native to S.A.)
Author Response
We appreciate the time and effort reviewers put into providing feedback on our manuscript, and we appreciate the insightful comments and valuable improvements to our paper. The reviewer’s comments and our responses are detailed below.
Comment: The manuscript was improved, but I believe in wrong direction. My comment was not taken to account. I thought it is a review paper on use of medicinal plants indigenous to Southern Africa for ZnO synthesis, but now, the paper is too explanatory on the use of general plants (not so limited to S.African).
Response: The authors strongly think that the Manuscript is still on track to cover the extensive work on indigenous medicinal plants of southern Africa that have been used in synthesising ZnO nanoparticles. All the medicinal plants (Agathosma betulina, Plumbago auriculata, Monsomia burkeana, Lessertia montana, Lessertia frutescens, Tulbaghia violacea, Aspalathus linearis, Dovyalis caffra, and Athrixia phylicoides DC) native to southern Africa that have been utilised for fabrication of ZnO nanoparticles are within the scope of the review paper.
Comment: I mentioned in my first review that authors should check and add other papers where ZnO was synthetised using native S. African (medicinal) plants. I do not understand why authors did not do this.
Response: We have done an extensive literature search and we found one additional medicinal plant that we had missed. 1. Athrixia phylicoides (Bush tea) which is medicinal plant that is indigenous to South Africa. DOI: 10.1021/acsomega.2c00530. 2. Dovyalis caffra (Kei apple) which is medicinal plant that is indigenous to South Africa. DOI: 10.1088/2053-1591/ab5bcb and DOI: 10.3390/molecules27103206.
Comment: I suggest that authors change the title or add more literature on the current title.
Response: The authors have declined the change of the title of the Manuscript and would like to humbly request the editor consider the title of the Manuscript as it is.
Comment: Here are some examples of papers that authors did not mention in the paper (and I believe there are more, but this would have to be searched by authors):
-https://doi.org/10.1016/j.reffit.2017.05.001 (Passiflora caerulea; native (indigenous) to S.Africa)
-https://doi.org/10.1007/s10904-022-02248-6 (Tulbaghia violacea; netive to S.A.)
-https://doi.org/10.3390/molecules27103206 (Dovyalis caffra; native to S.A.)
-https://doi.org/10.1088/2053-1591/ab1afa (Sutherlandia frutescens ; native to S.A.)
Response:
- Passiflora caerulea is native to Brazil and this is supported by this article https://www.ncbi.nlm.nih.gov/pmc/articles/PMC5963645/, therefore we could not add it as it does not fit with the scope of our work.
- Tulbaghia violacea has already been reviewed in the article. The reviewer might have missed it. Please refer to section 3.2.6 and line 506 of the article.
- Dovyalis caffra has been added.
- Sutherlandia frutescens has already been reviewed in the article under Lessertia frutescens. The reviewer might have missed it. Please refer to section 3.2.5 and line 460 of the article. We clearly state that Lessertia frutescens is formerly Sutherlandia frutescens.
Additionally, the authors have done an extensive search of any additional manuscripts that fit the scope of work and to the best of our knowledge there are no other south African plants that have utilized to fabricate ZnO nanoparticles.